# The Mechanical Performance and Reaction Mechanism of Slag-Based Organic–Inorganic Composite Geopolymers

**DOI:** 10.3390/ma17030734

**Published:** 2024-02-03

**Authors:** Xiaotong Xing, Weiting Xu, Guihua Zhang, Xilian Wen

**Affiliations:** 1School of Materials Science and Engineering, South China University of Technology, Guangzhou 510641, China; 202010103718@mail.scut.edu.cn (X.X.); zhangguihuascut@163.com (G.Z.); 2Guangzhou Residential Construction Development Co., Ltd., Guangzhou 510075, China; langer123456@126.com

**Keywords:** mechanical performance, reaction mechanism, slag-based geopolymer, organic–inorganic composite geopolymer, pressure-mixing process

## Abstract

A series of organic–inorganic composite geopolymer paste samples were prepared with slag-based geopolymer and three types of hydrophilic organic polymers, i.e., PVA, PAA, and CPAM, by ordinary molding and pressure-mixing processes. The reaction mechanism between slag-based geopolymer and organic polymers was studied by FT-IR, NMR, and SEM techniques. The experimental results showed that the slag-based geopolymer with the addition of 3% PVA presented the highest 28-day flexural strength of 19.0 MPa by means of a pressure-mixing process and drying curing conditions (80 °C, 24 h) compared with the geopolymers incorporating PAA and CPAM. A more homogeneous dispersion morphology was also observed by BSE and SEM for the 3% PVA-incorporated slag-based geopolymer. The FT-IR testing results confirmed the formation of a C–O–Si (Al) bond between PVA and the slag-based geopolymer. The deconvolution of the Q^3^ and Q^2^(1Al) species obtained by ^29^Si NMR testing manifested the addition of PVA and increased the length of the silicon backbone chain in the geopolymer. These findings confirmed that a composite geopolymer with high toughness can be produced based on the interpenetrating network structure formed between organic polymers and inorganic geopolymer.

## 1. Introduction

Slag-based geopolymer is one of the most promising new green cementitious materials due to its advantages of low energy consumption, small expansion, and strong corrosion resistance [1,2,3]. The active silico–aluminate materials are mixed with a strong alkaline solution such as an alkali metal (Na, K) hydroxide or silicate to synthesize geopolymer. In this environment, a three-dimensional network structure of geopolymers is formed due to the rapid dissolution and reunion reaction of silica–aluminum active substances. The geopolymer structure mainly contains three different sialate units [4], namely, poly sialate (–Si–O–Al–), poly sialate-siloxo (–Si–O–Al–O–Si–), and polysialate-disiloxo (–Si–O–Al–O–Si–O–Si–), shown in Figure 1. The properties of the final products are highly dependent on the cross-linking degree of the different silico–aluminate polymeric chains [5].

However, the low flexural strength and low fracture toughness of slag-based geopolymer limit its wider application in the construction field. It was found that when steel slag was used as the main aluminosilicate material to prepare geopolymer mortar, the flexural strength was only 11.7 Mpa while the compressive strength was 84 Mpa at 28 days [6]. In a study conducted by Kim et al. [7], the coefficient of flexural toughness was only 0.55 J. The defects of low toughness can be improved by making geopolymer composites. Many types of fillers such as various kinds of short and continuous fibers have been introduced into the geopolymer matrix to improve the mechanical properties [1,8,9]. Sükrü Özkan et al. [10] examined the effect of the hybrid use of 75% PVA fiber + 25% basalt fiber as additives on the mechanical performance of geopolymer concrete. The use of fiber in concrete significantly improved the resistance to the formation of cracks and contributed to increased ductility and the development of the mechanical strength and energy absorption of the concrete [11]. However, the reinforcement from a single species of fiber only physically improved the performance and crack resistance of the geopolymer matrix on one scale. The toughness of the geopolymer matrix itself was not improved. In addition, the compatibility of the fiber and geopolymer matrix also limited the construction and applications of the resulting composite materials.

It is noteworthy that a cement–polymer composite called “macro-defect-free (MDF) cement” with remarkably high toughness emerged in the early 1980s [12,13,14]. MDF cement is mainly composed of inorganic cement, water, and small amounts of organic polymer. The calcium aluminate cement–polyvinyl acetate polymer (CAC-PVAc) composite is one of the examples characterized by very high flexural strength of up to 70 MPa [15,16]. Based on micro-structural analysis, PVAc hydrolyzed under the high PH environment of the cement solution. Acetate ions dissolved from PVAc and subsequently reacted with the dissolved calcium ions from the calcium silicate in the cement to form calcium acetate [12,17] and, hence, formed an organic and inorganic bonding structure with high toughness.

The incorporation of various kinds of water-soluble polymers has been proven to improve the toughness and crack resistance of the cement and geopolymer matrix [18,19,20,21,22]. Catauroa [19] et al. introduced different proportions of polyethylene glycol (PEG) into metakaolin-based geopolymer and it was found that the PEG content and aging time affected the mechanical performance of geopolymers. The study also revealed that PEG leads to a network reorganization by increasing Al–O–Si bonds in the geopolymer matrix and forming H-bonds with the inorganic phase. Olesia Mikhailova [20] drew the conclusion that the maximum compressive and flexural strength of geopolymer is obtained with addition of 10% PEG400 because of the organic–inorganic bonding structure with high density and few pores. From the previous relevant studies, we have two points that need to be further studied and discussed.

First, the traditional common molding method may have a limited promotion effect on the bonding reaction of organic and inorganic compounds, and a better molding method needs to be explored. Second, the polymerization mechanism of water-soluble polymer and geopolymer needs more detailed and in-depth study.

The aim of this research is to investigate the influence of organic polymers with different functional groups (–OH, –COOH, and –CONH_2_) on the performances of slag-based geopolymers and select a suitable organic polymer with a sound modification effect on the toughness of slag-based geopolymer. In this study, we incorporated three types of water-soluble polymers, namely, PVA, PAA, and CPAM, as organic additives into slag-based geopolymer to investigate the mechanical properties and polymerization reaction mechanism of the organic–inorganic hybrid composite geopolymer. To strengthen the organic–inorganic bonding reaction, the pressure-mixing process was adopted and compared to the traditional common modeling method in the process of making geopolymer paste specimens.

## 2. Materials and Methods

### 2.1. Materials

The slag used in this study was the S95 mine powder from Ma’anshan Tianrui slag trading Co., Ltd., Ma’anshan, China. The chemical composition of the slag examined by XRF is shown in Table 1. The particle size of the slag was analyzed by a laser particle size meter, and the particle size distribution is shown in Figure 2.

Based on the oxide components of slag in Table 1, the correlation quality indicators of the slag were calculated by the following Formulas (1)–(4) [23,24]. According to the calculation results, the slag was evaluated as being of a high-quality grade.
(1)$$b=\frac{\mathrm{CaO}+\mathrm{MgO}+{\mathrm{Al}}_{2}O_{3}}{{\mathrm{SiO}}_{2}}=\frac{37.12+10.77+14.84}{32.06}=1.96>1$$
(2)$$H_{0}=\frac{{\mathrm{Al}}_{2}O_{3}}{{\mathrm{SiO}}_{2}}=\frac{14.84}{32.06}=0.46>0.25$$
(3)$$M_{0}=\frac{\mathrm{CaO}+\mathrm{MgO}}{{\mathrm{SiO}}_{2}+{\mathrm{Al}}_{2}O_{3}}=\frac{37.12+10.77}{32.06+14.84}=1.02>1$$
(4)$$K=\frac{\mathrm{CaO}+\mathrm{MgO}+{\mathrm{Al}}_{2}O_{3}}{{\mathrm{TiO}}_{2}+\mathrm{MnO}+{\mathrm{SiO}}_{2}}=\frac{37.12+10.77+14.84}{32.06+0.33+0.92}=1.88>1.2$$

The alkali activator used in this study was water glass (industrial-grade sodium silicate) provided by Foshan Shengjing New Material Company (Foshan, China). The modulus of the water glass was 2.44 and the solid content was 46.17%, with 13.70% Na_2_O and 32.47% SiO_2_. Sodium hydroxide (analytical purity, Tianjin Jinhui Taiya Chemical Reagent Co., Ltd., Tianjin, China) was adopted to adjust the modulus of the alkaline activator.

The water-soluble organic polymers used in this study were cation polyacrylamides (CPAM), polyvinyl alcohol (PVA), and polyacrylic acid (PAA). They were purchased from Shanghai Macklin Chemical Reagent Co., Ltd. (Shanghai, China). Both PVA and CPAM were granular. During the test, they were respectively pre-mixed with water to form a solution with 55% concentration for subsequent experiments. PAA was a liquid with a concentration of 50%.

### 2.2. Mixing Proportions

Before preparing organic and inorganic composite geopolymers, to achieve a suitable slag-based geopolymer with high mechanical performance as the inorganic substrate, the effects of the type, modulus, and concentration of alkali activators on the strength and polymerization degree of slag-based geopolymer are investigated in Section 3.1.

The mixing proportions of organic–inorganic composite geopolymer pastes are given in Table 2. It is worth pointing out that the proportions of water-soluble polymers added were 0.5%, 1%, and 2% for the common molding method and 1%, 3%, and 5% for the pressure-mixing process.

### 2.3. Ordinary Molding Process

The water-to-solid ratio of the geopolymer pastes was 0.35. All geopolymer pastes were mixed in a Hobart mixer at an ambient temperature. The alkaline solution was prepared by an alkali activator and water in advance. Then, the slag was added to the alkaline solution and stirred for 4 min. Finally, the water-soluble polymer was added into the geopolymer slurry and slowly stirred for 6 min. All pastes were cast with sizes of 25 mm × 25 mm × 140 mm and 20 mm × 20 mm × 20 mm for flexural strength and compressive strength tests, respectively. For the first 24 h, the pastes were sealed with a layer of polyethylene to prevent the evaporation of moisture. Then, they were demolded, tightly sealed in plastic bags, and cured in a steam-curing chamber with a temperature of 20 ± 1 °C and 99% relative humidity until the testing dates.

### 2.4. Pressure-Mixing Process

To promote the breaking and repolymerization of more chemical bonds in organic molecules and aluminosilicates in geopolymers, the pressure-mixing process was used. The mixer parameter was set to the front and rear roller roll ratio of 1:3, and the roll distance was 1.5 mm. The plastic organic–inorganic composite geopolymer slurry was subjected to mixing for 5 min. The obtained mixture was combined by passing it repeatedly through the nip of two steel roller mills at narrow nip gaps and, subsequently, a higher shearing was applied to the mixture with higher rotating speeds. A paste sheet was laminated in a mold with a size of 120 mm × 30 mm × 10 mm and molded by a universal testing machine. In addition, to explore the influence of curing conditions on strength, the pastes were sealed by a layer of polypropylene after being demolded for 24 h. Then, half of the pastes were placed in a steam-curing chamber with 99% relative humidity at 80 °C for 24 h, whereas the remaining pastes were placed in an oven and cured at 80 °C for 24 h.

### 2.5. Mechanical Properties of the Pastes

Pastes were cast into 25 mm × 25 mm × 140 mm, 10 mm × 30 mm × 120 mm, and 20 mm × 20 mm × 20 mm molds for mechanical property tests [25]. The flexural and compressive strength of the pastes was measured by a fully automatic universal testing machine at the ages of 3 d, 7 d, and 28 d. Flexural strength tests were carried out using the three-point bending method at a loading rate of 2 mm/min. The strength result was based on the average of three testing pastes for each mix.

### 2.6. The Characterization of Organic–Inorganic Composite Material

#### 2.6.1. Fourier Transform Infrared Reflection (FTIR) Tests

FT-IR (Vertex 70 produced by Bruker, Ettlingen, Germany) tests were conducted in a transmission model with a scan range of 4000–400 cm^−1^. The KBr tablet method was used in the test. The ratio of the powder sample to dry KBr was taken from 1:100 to 1:200 in an agate mortar for 5 min, and the powder was put into a grinding mixture after no reflection. The piezoelectric sheet was pressed internally and, finally, the transparent circular sheet was tested by infrared spectra.

#### 2.6.2. Solid-State ^29^Si Nuclear Magnetic Resonance (NMR) Tests

Avance III PULS 400 MHz NMR equipment, produced by Bruker, Germany, was used to identify and quantify different types of silicon. The resonance frequency was 79.3 mHz, the rotation angle was π/2, the pulse width was 4 μs, and the magic angle rotation frequency was 8 kHz. In addition, PeakFit 4.0 software was used for the peak fitting of the original map for data processing in this experiment. To accurately separate peak results, the AutoFit Peaks III Deconvolution program was used, and the relevant parameter r2 was greater than 0.99.

#### 2.6.3. Inductively Coupled Plasma Optical Emission Spectrometry (ICP-OES) Tests

The concentrations of Si, Ca, and Al ions in the solution were determined by the ICP-OES method. The mixing ratio of the test solution is shown in Table 3. At 30 min, 60 min, 90 min, 120 min, and 180 min, with centrifugation at 3000 rpm for 10 min, the solution was separated and collected from the paste, and the clean solution was obtained after filtration. The plasma gas flow rate was 15 L/min. The atomized gas flow rate was 0.6 L/min. The pump speed was 100 rpm.

#### 2.6.4. Backscattered Electron (BSE) and Scanning Electron Microscope (SEM) Tests

The morphological characteristics and element distribution of the geopolymer pastes were assessed using SEM (SU8220, HITACHI, Tokyo, Japan) with BSE. The working distance was 12 mm and the voltage was 10 kV under the secondary electron (SE) mode.

#### 2.6.5. Mercury Intrusion Porosimetry (MIP) Tests

The pore size distribution of the pastes was determined using the MIP method with the Micromeritics Auto Pore IV 9500 mercury injection instrument (Micromeritics Instrument Corporation, Norcross, GA, USA). The test pressure range of the MIP was 0.0036–210 MPa.

## 3. Results and Discussion

### 3.1. The Effects of Different Alkali Contents and Moduli on the Properties of Slag-Based Geopolymer

#### 3.1.1. Mechanical Properties of Pastes

The compressive strength and the flexural strength of slag-based geopolymer pastes with different alkali activator (sodium hydroxide) contents are shown in Figure 3. The highest compressive and flexural strength was observed for pastes with 6% alkali content.

For the geopolymer with water glass as the alkali activator, the effect of the modulus of water glass on the compressive and flexural strength of the pastes is shown in Figure 4. It can be observed that pastes with a modulus of 1.6 and 6% alkali content presented the highest compressive strength of 91.0 Mpa and flexural strength of 7.5 Mpa at 28 d, respectively. Comparing the results of Figure 3 and Figure 4, it is more appropriate to choose water glass as the activator for geopolymer pastes.

#### 3.1.2. ICP Analysis

The evolution of Si^4+^, Ca^2+^, and Al^3+^ concentrations in the solutions of slag pastes with different alkali contents is shown in Figure 5. As the alkali content increased, the high dissolution concentration of Si^4+^ reduced over time. This was caused by more OH^−^ and Na^+^ in the solution, which were available in the network structure of the vitreous body. Then, Si^4+^ gradually formed [SiO_4_]^4−^ combined with [AlO_4_]^5−^ and Ca^2+^. Ca^2+^ and Al^3+^ concentrations with a 6% alkali content were extremely high, which was beneficial for the polymerization reaction. The dissolution–polymerization process equilibrated to form more hydration products. The concentration changes of Si^4+^, Ca^2+^, and Al^3+^ in the slag paste solutions with different alkali contents over time are shown in Figure 5. The dissolved concentration of Si^4+^ decreased with the extension of time and increased with the increase in alkali content. This was due to the fact that more OH^−^, Ca^2+^, and Al^3+^ dissolved in the solution were available in the reticular structure that formed the vitreous body. Si^4+^ then gradually formed [SiO_4_]^4−^ and [AlO_4_]^5−^, which bonded with Ca^2+^. When the alkali content was 6%, the concentrations of Ca^2+^ and Al^3+^ in the solution reached the highest values, which was conducive to promoting the polymerization reaction and forming more polymerization products.

#### 3.1.3. FTIR Analysis

To study the effect of alkali concentration on the chemical bond of slag-based geopolymer, paste samples were subjected to FTIR analysis. The results are shown in Figure 6. At 1003~1035 cm^−1^, there was a large diffuse band that lacked periodicity, representing the characteristic of amorphous Si-Al polytetrahedral. For the geopolymers with 6% alkali content, the Si-O-Si peaks at 460 cm^−1^ and the Si-O-Al peaks at 960 cm^−1^ were sharper and more intense. This can be attributed to more polycondensation occurring between [SiO_4_]^4−^ and [AlO_4_]^5−^ [26]. In addition, geopolymers with 6% alkali content were found to have more strength at 891 cm^−1^, 1021 cm^−1^, and 1097 cm^−1^, representing the diverse characteristic groups of C-(A)-S-H polymerization products (SiO_4_)^4−^, (SiO_3_)^2−^, and SiO_2_, respectively. It is apparent that an alkali content of 6% is the most favorable alkaline environment to promote the polymerization of slag-based geopolymer [27].

#### 3.1.4. ^29^Si NMR Analysis

The polymeric structure of geopolymer was studied by the ^29^Si nuclear magnetic resonance technique. The distribution of the polymerization degree and main chain length of slag-based polymers prepared with different alkali contents and different moduli of activators are shown in Table 4 and Table 5. The polymerization degree analysis showed that the cumulative strength of Q^0^ was the smallest when the alkali content was 6%, which indicated that more slag was consumed. The average chain length of C-(A)-S-H in the hydration product could be calculated by Equation (5) [28]. When the base content was 6% and the modulus was 1.6, the average chain length was the largest.
(5)$$MCL=\frac{I\left(Q^{1}\left(0Al\right)\right)+I\left(Q^{2}\left(0Al\right)\right)+\frac{3}{2}I\left(Q^{2}\left(1Al\right)\right)}{\frac{1}{2}I\left(Q^{1}\left(0Al\right)\right)}$$

### 3.2. Effect of Water-Soluble Polymer Type on Slag-Based Geopolymer Paste

#### 3.2.1. Mechanical Properties

The compressive strength and the flexural strength of slag-based geopolymer with three water-soluble polymers at the age of 3 d, 7 d, and 28 d are shown in Table 6. For all the mixtures, 6% water glass with a modulus of 1.6 was used as the alkali activator. PVA-0.5 showed the highest compressive strength and flexural strength among all the pastes. Under the same curing conditions, the PAA-Na and CPAM pastes presented lower flexural strength results than those of the reference mortar.

#### 3.2.2. FTIR Analysis

FTIR spectra of slag-based geopolymer composite pastes without and with the incorporation of 0.5%, 1%, and 2% PVA at 28 d are shown in Figure 7. The absorption peaks at 452 cm^−1^ representing the Si-O bending vibration bond gradually moved to the higher wave number with increases in the PVA dosage, indicated by the paste with 0.5% PVA (460 cm^−1^), the paste with 1% PVA (475 cm^−1^), and the paste with 2% PVA (489 cm^−1^). The results show that the addition of PVA led to an increase in the Si–O tetrahedral polymerization structure in C–S–H [29,30,31].

The stretching modes of the SiO_n_ polymers and monomers with n = 4, 3, 2, 1, and 0 in geopolymer corresponded to the absorption bands at 1200 cm^−1^, 1100 cm^−1^, 950 cm^−1^, 900 cm^−1^, and 850 cm^−1^, respectively. The silicon of SiO_n_ is prone to being replaced by aluminum due to the weak Al–O bond. With n from 4 to 0, the expansion pattern of the absorption band at 1200 cm^−1^, 1100 cm^−1^, 950 cm^−1^, 900 cm^−1^, and 850 cm^−1^ shifted to a lower wave number. It is worth pointing out that the absorption peak at 986 cm^−1^ was caused by the asymmetric tensile vibration of the Si–O–Si (Al) bond belonging to the characteristic C–Write in formal instead of italics Write in formal instead of italics (A)–S–H [32]. With the increase in the PVA dosage, the absorption peak gradually moved towards the lower wave number, indicating that the addition of PVA promoted the binding of [SiO_4_]^4−^ and [AlO_4_]^5−^.

The vibration peak of –OH at 2933 cm^−1^ in PVA was overlapped by Si–OH in the polymerizate. The absorption peak at 3463 cm^−1^ was attributed to the asymmetric stretching vibration of the hydroxyl group [33], which broadened with the increase in PVA addition.

It is worth noting that when the PVA dosage was 2.0%, there was a weak absorption peak at 1180 cm^−1^ attributed to the C–O–Si (Al) vibration [34] formed by the organic–inorganic chemical reaction.

#### 3.2.3. ^29^Si NMR Analysis

The ^29^Si NMR spectra of the reference slag-based geopolymer paste and the geopolymer paste with 0.5% PVA are shown in Figure 8. The polymerization distribution, main chain length, and reaction degree of polymerization of the [SiO_4_]^4−^ tetrahedra in the paste at 28 d were calculated based on the results in Figure 8 and are shown in Table 7.

Compared with the reference paste, the paste with 0.5% PVA content did not detect the Q^3^(1Al) silica tetrahedron, which may have been due to PVA preventing Al atoms from replacing Si atoms in the Q^3^ silica tetrahedron or PVA grafting at a missing bridge silicon-tetrahedron site to form C–O–Si bonds. From the MCL results, the average chain length with 0.5% PVA was longer, and the Q^1^ (0Al) and Q^2^ (1Al) silicone tetrahedrons had a higher accumulation intensity, which indicated the presence of more chain structures in this paste. However, the accumulation intensity of the Q^3^ (0Al) silicone tetrahedron was only 0.87%, which indicates that it did not easily form a high polymerization product with the addition of the PVA. However, according to reference [35], it is known that the main polymerization structure of influential properties is Q^1^, Q^2^, so the 0.5% PVA sample had better mechanical properties.

The reaction degree (α) of the geopolymer was estimated according to Equation (6). *I* [Q^0^] is proportional to the number of unreacted phases and I [Q^1^] and I [Q^2^] are proportional to the number of reaction products. From the calculation results shown in Table 7, the reaction degree reactivity of the reference paste was 82.85%, while that of 0.5% PVA was 73.14%. It is possible that the introduction of PVA reduced the degree of polymerization of aluminosilicate in the slag.
(6)$$\alpha =\left[1-\frac{I\left(Q^{0}\right)}{\sum_{i=0}^{2}I\left(Q^{2}\right)}\right]*100\%$$

#### 3.2.4. SEM Analysis

The morphologies of the reference geopolymer paste, geopolymer paste with 0.5% PVA, and geopolymer paste with 2.0% PVA at 28 d are shown in Figure 9. It apparent that the hydration product of the reference geopolymer paste was relatively large. The hydration products of geopolymer paste containing 0.5% PVA exhibited a scale-like structure, and the bonds between scales were very tight to form a whole structure. When the amount of PVA in the geopolymer was increased to 2.0%, the hydration products formed thin sheets and the geopolymer matrix turned fluffy. It is apparent that the excessive addition of PAV may lead to a decrease in the homogeneity and mechanical strength of the matrix.

### 3.3. The Effect of the Pressure-Mixing Process of Pastes with Various Dosages of PVA under Different Curing Conditions

#### 3.3.1. The Mechanical Properties of Pastes with Various Dosages of PVA Prepared by the Pressure-Mixing Process

The flexural strength of geopolymer pastes with 1%, 3%, and 5% PVA content obtained by the pressure-mixing process is shown in Figure 10. The incorporation of 3% PVA noticeably increased the flexural strength of pastes, with maximum values of 10.9 MPa at 3 d, 11.3 MPa at 7 d, and 12.6 MPa at 28 d, respectively. However, a further increase in the PVA dosage to 5.0% did not further improve the strength. An excessive increase in the amount of PVA led to the local agglomeration of organic polymer and reduced the effective site of the organic–inorganic polymerization reaction, which led to the weakening of organic–inorganic composite geopolymer. Thus, the dosage of 3% PVA was used for subsequent experiments. In contrast, using ordinary processes under the same curing conditions, the flexural strength began to decrease when the PVA content was increased by 1% or more. These results indicate that the pressure-mixing process significantly improves the flexural strength.

#### 3.3.2. Mechanical Properties of Pastes with 3% PVA under Different Curing Conditions

The flexural strength of pastes with 3% PVA under different curing conditions are shown in Figure 11. Compared with the pastes cured under normal conditions (20 °C, 24 h), the flexural strengths of pastes under dry curing conditions (80 °C oven maintenance, 24 h) and steaming curing conditions (80 °C teaming maintenance, 24 h) were generally enhanced, especially for the ones under 80 °C dry conditions. Thus, dry curing is recommended for geopolymer pastes containing water-soluble polymers.

#### 3.3.3. FTIR Analysis

The FTIR spectra of geopolymer pastes with different PVA dosages prepared by the pressure-mixing process with different curing regimes at 28 d are shown in Figure 12. As seen in Figure 12a, with an increase in the addition of PVA, the absorption peak of pastes at 473 cm^−1^ shifted slightly to the left, indicating that the degree of polymerization between the Si–O bonds decreased. The tensile vibration peak of the –OH bond was mainly at 3522 cm^−1^, and the smaller absorption peak at 1655 cm^−1^ was the bending vibration peak of H–O–H. With an increase in PVA content, the peak position of both sites shifted to the right because the hydrogen bond formed by the binding water of the hydration product was reduced. Compared with the normal molding process, the characteristic peak of C–A–S–H near 1006 cm^−1^ was removed by about 50 wave numbers, indicating the formation of a high polymerization structure. According to Figure 12b, the main absorption peaks of pastes under different curing regimes in the spectra were basically consistent. However, the wave numbers of the main absorption peaks were obtained at 3498 cm^−1^, and the peak shape was widened for pastes under steaming curing conditions, indicating an increase in binding water content and hydrogen bonds.

#### 3.3.4. ^29^Si NMR Analysis of Pastes with Different Dosages of PVA Prepared by the Pressure-Mixing Process under Different Curing Regimes

The ^29^Si NMR spectra of pastes prepared by the pressure-mixing process with 1% PVA under normal curing conditions, 3% PVA under normal curing conditions, 5% PVA under normal curing conditions, 3% PVA under dry heat curing conditions, and 3% PVA under steam curing conditions at 28 d are shown in Figure 13a–e, respectively. The polymerization distribution of pastes prepared by the pressure-mixing process under different curing regimes at 28 d are given in Table 8. Compared with Figure 13b,d,e, when the PVA content was 3%, the original spectrum of the high-temperature cured paste was sharper and less smooth. The position of the strongest peak gradually shifted to the lower direction of the magnetic field. Combined with the results in Table 8, it is apparent that Si in the Q^2^ silico-oxygen tetrahedron in the paste was more easily replaced by a certain amount of Al atoms. The cumulative strength of Q^3^ silica tetrahedrons cured at high temperatures was twice that at normal temperatures, indicating that high-temperature conditions promote the polycondensation reaction between silica tetrahedrons, which is conducive to the formation of a one-layer or double-chain high-polymerization structure.

#### 3.3.5. Morphology Analysis

##### BSE Analysis

The BSE images of pastes prepared by the pressure mixing process with 1% PVA under normal curing conditions, 3% PVA under normal curing conditions, 5% PVA under normal curing conditions, 3% PVA under dry heat curing conditions, and 3% PVA under steam curing conditions at 28 d are shown in Figure 14a–e, respectively. Compared with Figure 14a–c, it can be observed that when the PVA dosage was 1%, no organic phase was found, while an organic phase was observed in the paste with a 5% PVA dosage, as shown in Figure 14c. For the pastes under normal curing conditions, with an increase in PVA content, cracks in the geopolymer matrix became obvious, maybe due to the different hardening rates of organic particles and slag-based geopolymer. Compared with Figure 14b,d,e, the paste under high-temperature curing had a relatively dense structure, and there was no obvious crack around the organic phase, but the dispersion was still uneven. In addition, by comparing Figure 14d,e, it can be observed that different high-temperature curing regimes had different effects on the PVA state in the matrix. With dry heat curing, PVA bonded more with slag geopolymer, and there was no obvious two-phase interface. For the paste under steaming curing conditions, the interface between the embedded PVA and the slag geopolymer matrix was obvious.

##### SEM Analysis

The morphology images of pastes prepared by the pressure mixing process with 1% PVA under normal curing conditions, 3% PVA under normal curing conditions, 5% PVA under normal curing conditions, 3% PVA under dry heat curing conditions, and 3% PVA under steam curing conditions at 28 d are shown in Figure 15a–e, respectively. As shown in Figure 15a–c, with an increase in PVA dosage, no characteristic reaction product appeared, and the geopolymer matrix had a uniform texture, indicating that a good bond formed between the organic and inorganic phases under the pressure-mixing process. Compared with Figure 15b,d,e, the reaction product under different curing conditions had a different morphology. For the paste under dry curing conditions, there was a characteristic morphology of the internal structure of the pomegranate. This was because the PVA, which had lost water, wrapped some slag particles and rapidly shrunk under dry curing conditions. This kind of organic–inorganic composite structure is closely combined with the hydration product so that the material is not prone to brittle cracks during the loading process. For the paste under steam curing conditions, a variety of reaction products were produced in the dense matrix due to the early water retention effect.

#### 3.3.6. Porosity and Pore Size Distribution of Pastes

The pore size distribution and cumulative pore volume of hardened pastes with 3% PVA prepared by different molding processes and curing regimes at 28 d are shown in Figure 16. Figure 16a shows that the order of the maximum pore size of each sample is paste prepared by ordinary molding (60.77 μm) > paste prepared by mixing molding > paste prepared by pressure mixing molding. The results show that the pressure mixing process eliminated some large pores in the paste matrix. However, the cumulative pore volume of each paste was contrary to the expected results, which may indicate that some pore defects were introduced during the pressure mixing process itself, resulting in an increase in the total pore volume of the paste.

It is seen from Figure 16 that the order of maximum pore size and cumulative pore volume of each paste is as follows: paste under room temperature curing sample > paste under dry heat curing sample > paste under steam curing sample. The results show that under the high-temperature curing conditions, the reactivity of slag geopolymer was enhanced. More gel products were produced in the system, and the voids were gradually filled by slag particles, which reduced the pore size and made the matrix compact.

The maximum pore size and total pore volume of the pastes treated with steam curing were 50.34 nm and 0.0443 mL/g, respectively, which were the smallest among all the pastes. The reason for this is that steam curing continued to provide water for the unreacted slag particles in the later stage of the geopolymer reaction, resulting in more reaction products to fill the pores, refine the pores, and reduce the total pore volume of the paste. These results explain the reason for the maximum bending strength of the paste under steam curing conditions.

## 4. Conclusions

The effect of water-soluble polymers as additives on the mechanical and structural properties of slag-based geopolymer was investigated in this study. To reveal the modification mechanism of organic additives on slag-based geopolymer, the pore diameter distribution, boning, and polymerization degree of organic–inorganic composite geopolymer were explored. Based on this study, the following conclusions can be drawn:For geopolymer paste using sodium silicate as an alkali activator (alkali content of 6% and modulus of 1.6) prepared by the ordinary production process under the same curing age, the flexural strength of the geopolymer paste with 0.5 wt% PVA was 9.2 Mpa, which was 29.33% higher than that of the reference geopolymer paste.By means of the pressure-mixing molding process, the fraction of large capillary pores in the geopolymer pastes was significantly reduced. The flexural strength of the paste with 3% PVA was 12.6 Mpa at 28 d, while the flexural strength value increased markedly to 19.0 MPa due to the acceleration of the polymerization reaction by dry curing (80 °C, 24 h).The FTIR spectrum of 2% PVA modified slag-based geopolymer shows that the vibration peak near 1180 cm^−1^ was due to the presence of a C–O–Si bond. The results show that the -OH functional group in C–(A)–S–H reacted with the –OH in PVA, which provided the possibility for the formation of an organic–inorganic interpenetration network. The ^29^Si NMR spectra demonstrated that the polymerization degree of the aluminosilicate and the chemical environment around the [SiO_4_]^4−^ tetrahedra were affected by the introduction of PAV into the geopolymer matrix. An appropriate PVA content incorporated into slag-based geopolymer led to the increased generation of layered-group (Q^3^) tetrahedra, indicating an increase in the polymerization degree of aluminosilicate tetrahedra.BSE analysis showed that PVA uniformly dispersed in the inorganic matrix without an obvious interface under dry curing conditions. Due to the influence of the pressure-mixing process, the reaction product bound more closely to the unreacted slag.

## Figures and Tables

**Figure 1 materials-17-00734-f001:**
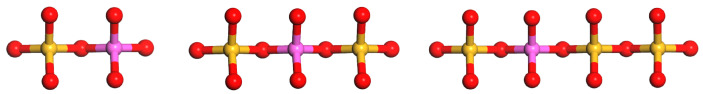
Structural unit model of geopolymer. (The red ball represents oxygen atom, the yellow ball represents the silica atom, and the purple ball represents the aluminum atom).

**Figure 2 materials-17-00734-f002:**
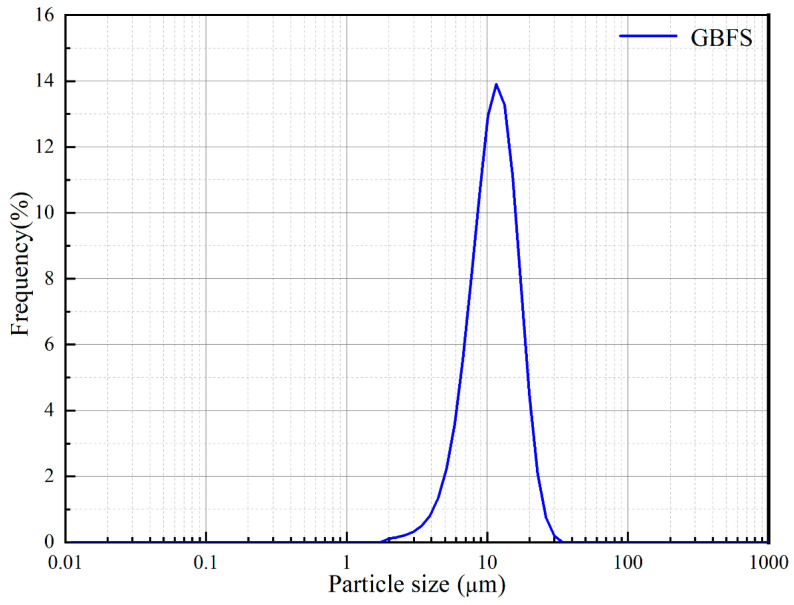
Particle size distributions of slag.

**Figure 3 materials-17-00734-f003:**
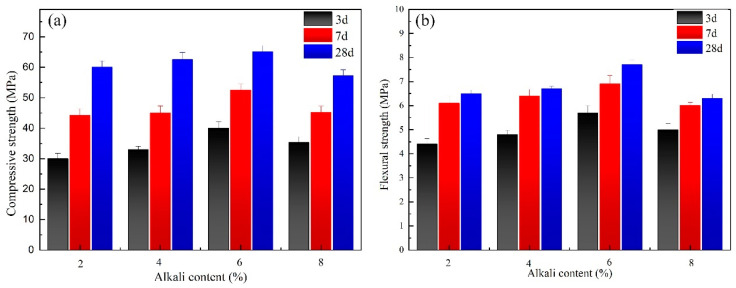
Effect of alkali content on mechanical properties of slag-based geopolymer paste: (**a**) compressive strength; (**b**) flexural strength.

**Figure 4 materials-17-00734-f004:**
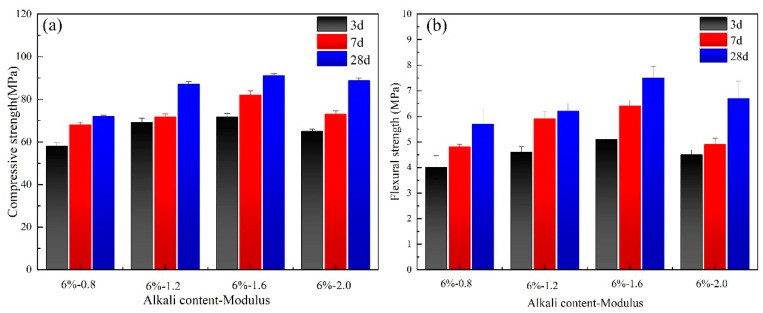
Effect of modulus on mechanical properties of slag-based geopolymer: (**a**) compressive strength; (**b**) flexural strength.

**Figure 5 materials-17-00734-f005:**
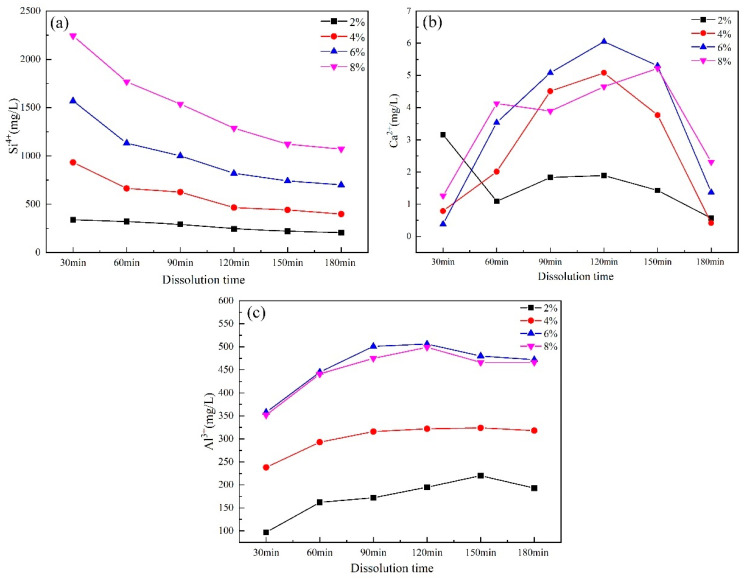
Ion concentration of slag in slurry with large water–slag ratios and different alkali contents: (**a**) Si^4+^; (**b**) Ca^2+^; (**c**) Al^3+^.

**Figure 6 materials-17-00734-f006:**
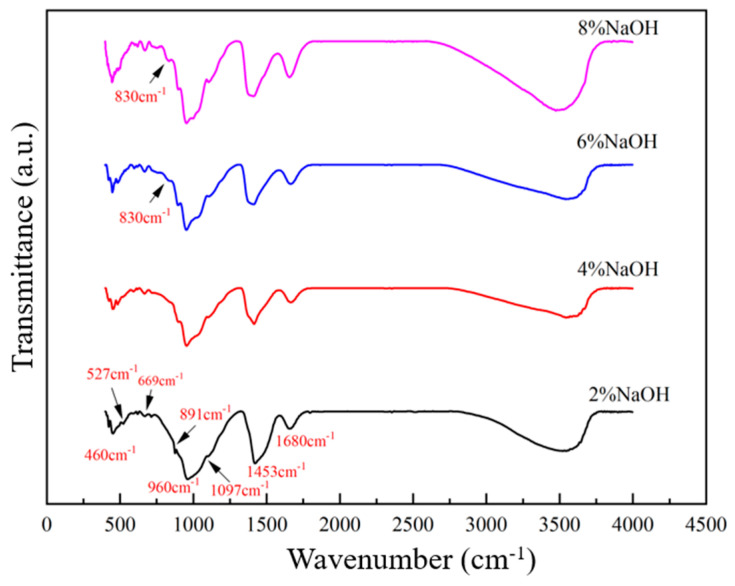
FTIR spectra of slag-based geopolymers prepared with NaOH activators at 28 d with different alkali contents.

**Figure 7 materials-17-00734-f007:**
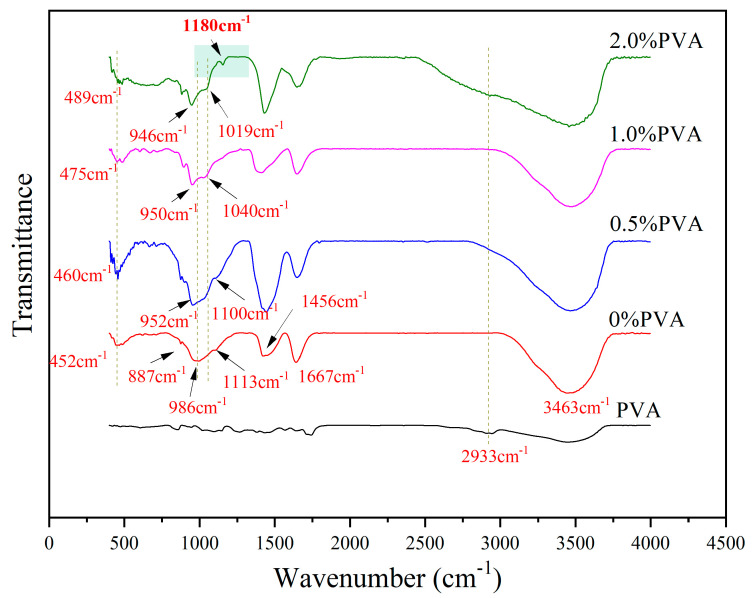
FTIR spectra of paste at 28 d.

**Figure 8 materials-17-00734-f008:**
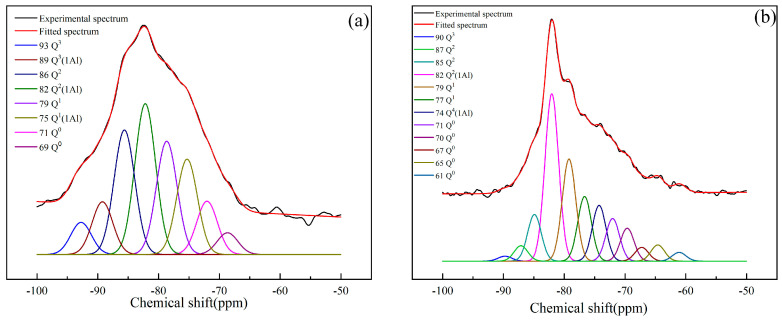
^29^Si NMR spectra of (**a**) reference geopolymer paste and (**b**) geopolymer paste with 0.5% PVA.

**Figure 9 materials-17-00734-f009:**
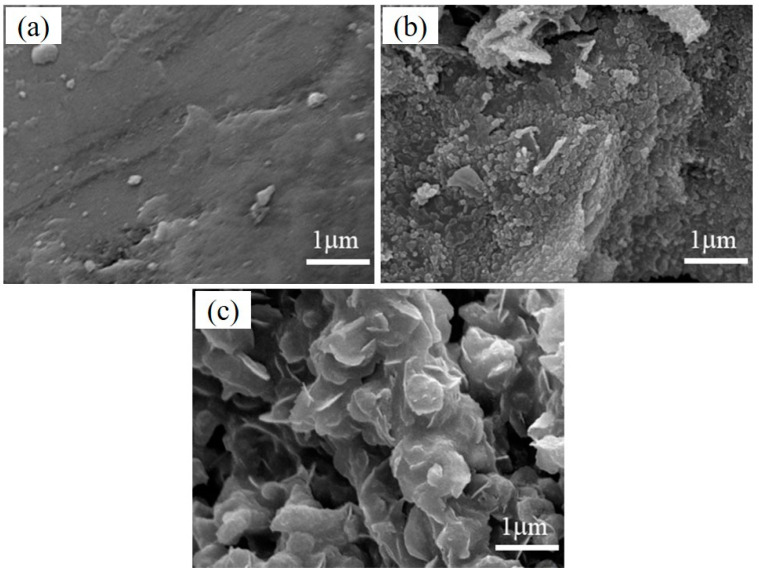
Morphology of (**a**) reference geopolymer paste, (**b**) geopolymer paste with 0.5% PVA, and (**c**) geopolymer paste with 2.0% PVA at 28 d.

**Figure 10 materials-17-00734-f010:**
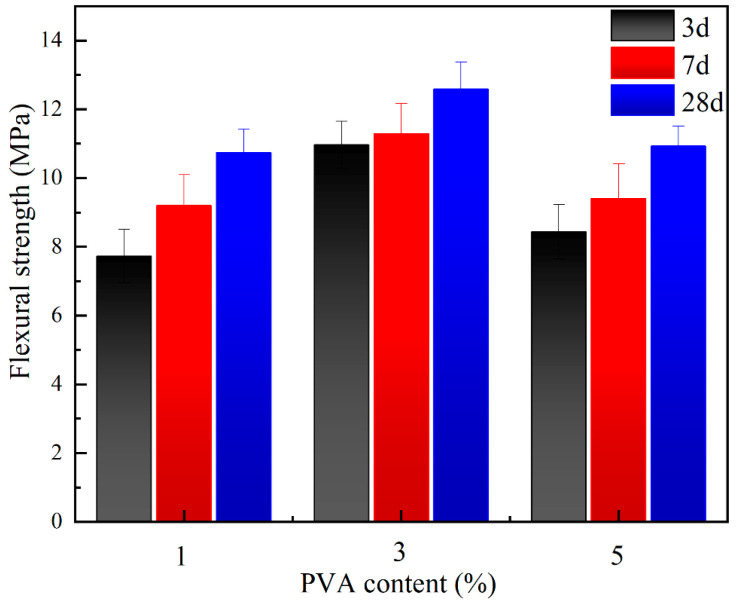
Flexural strength of pastes prepared by pressure-mixing process.

**Figure 11 materials-17-00734-f011:**
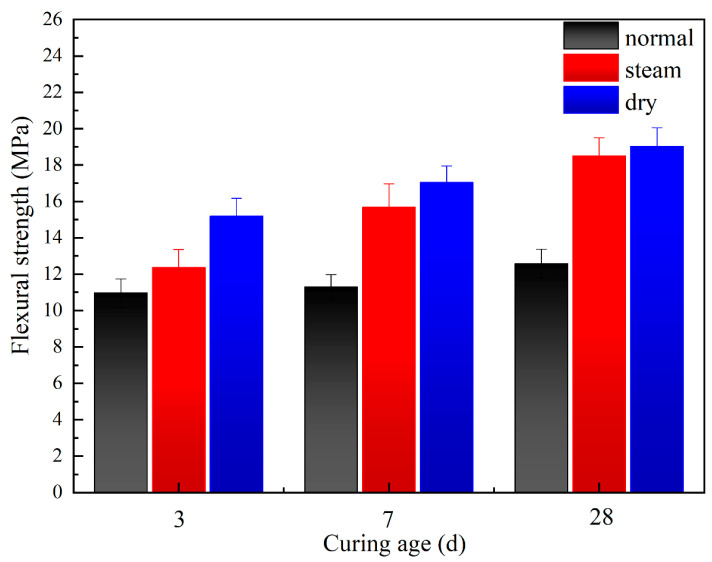
Flexural strength of pastes prepared by different curing conditions.

**Figure 12 materials-17-00734-f012:**
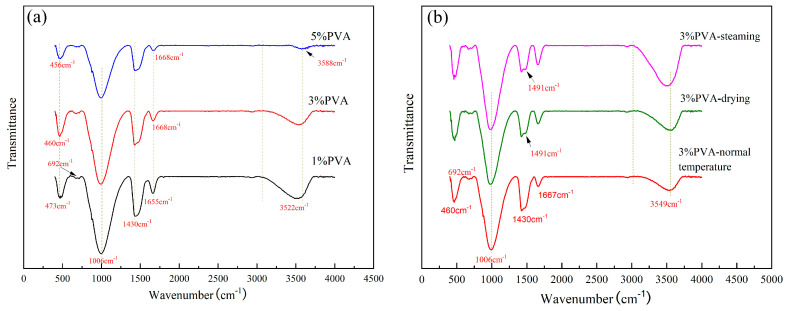
FTIR spectra of pastes prepared by pressure-mixing process with (**a**) different contents of PVA and (**b**) different curing regimes at 28 d.

**Figure 13 materials-17-00734-f013:**
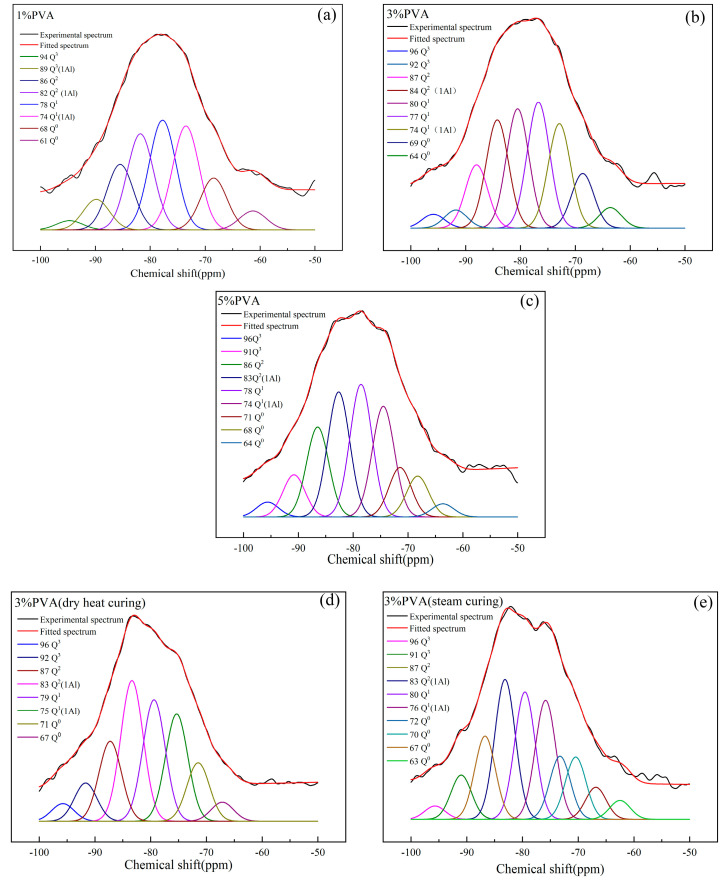
^29^Si NMR spectra of pastes prepared by pressure-mixing process with (**a**) 1% PVA (normal curing), (**b**) 3% PVA (normal curing), (**c**) 5% PVA (normal curing), (**d**) 3% PVA (dry heat curing), and (**e**) 3% PVA (steam curing) at 28 d.

**Figure 14 materials-17-00734-f014:**
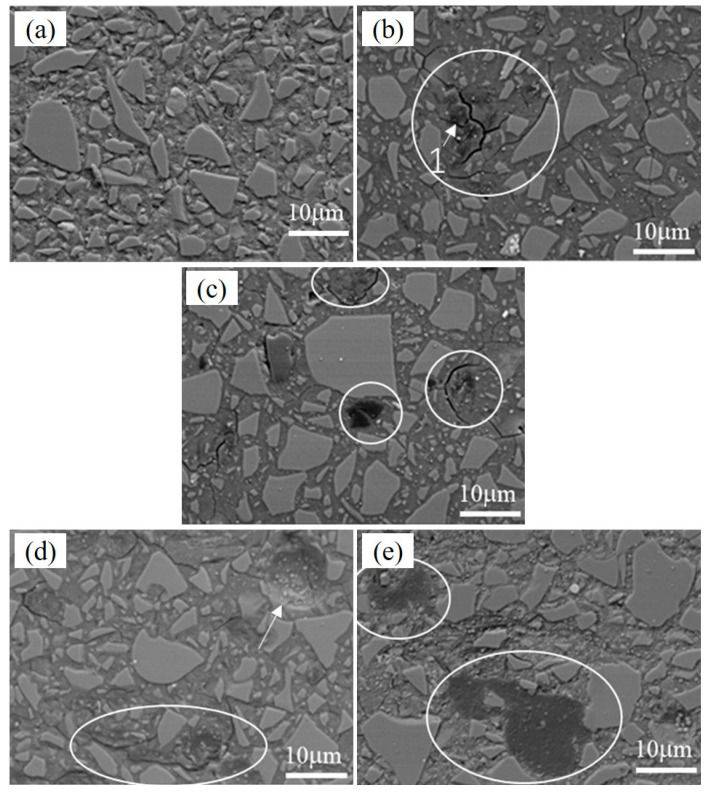
BSE images of pastes prepared by pressure mixing process with (**a**) 1% PVA (normal curing), (**b**) 3% PVA (normal curing), (**c**) 5% PVA (normal curing), (**d**) 3% PVA (dry heat curing), and (**e**) 3% PVA (steam curing) at 28 d.

**Figure 15 materials-17-00734-f015:**
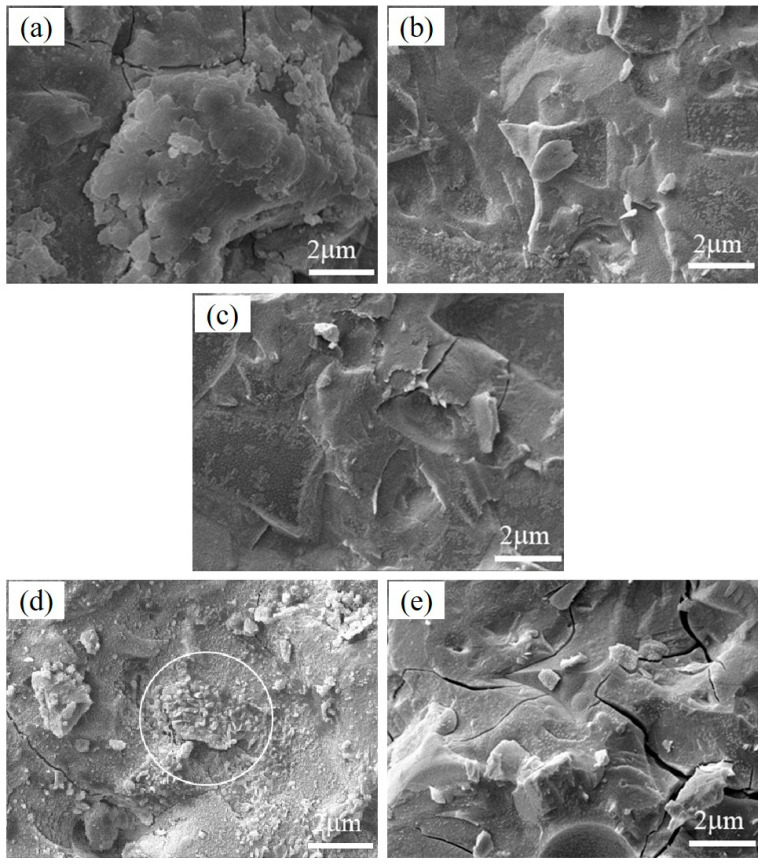
Morphology of pastes prepared by pressure mixing process with (**a**) 1% PVA (normal curing), (**b**) 3% PVA (normal curing), (**c**) 5% PVA (normal curing), (**d**) 3% PVA (dry heat curing), and (**e**) 3% PVA (steam curing) at 28 d.

**Figure 16 materials-17-00734-f016:**
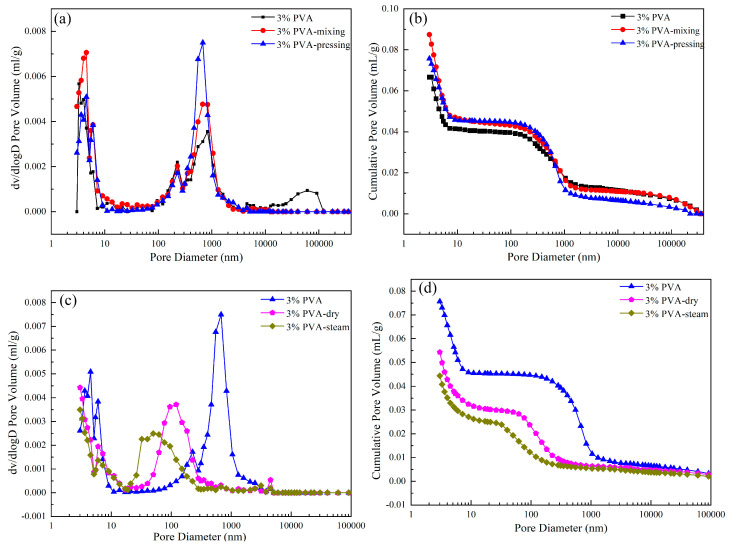
(**a**) Pore size distribution of pastes prepared by different molding processes at 28 d, (**b**) cumulative pore volume of pastes prepared by different molding processes at 28 d, (**c**) pore size distribution of pastes prepared by different curing regimes at 28 d, (**d**) cumulative pore volume of pastes prepared by different curing regimes at 28 d.

**Table 1 materials-17-00734-t001:** Chemical compositions of slag.

Components (%)	CaO	SiO_2_	Al_2_O_3_	MgO	SO_3_	Fe_2_O_3_	TiO_2_	MnO	LOI
slag	37.12	32.06	14.84	10.77	1.67	1.14	0.92	0.33	1.07

Note: LOI: Loss of ignition.

**Table 2 materials-17-00734-t002:** Mixture proportions of pastes in weight (%).

Mix ID	Polymer Type		Mix Proportions			
	Species	Weight	Water	Slag	BaCl_2_	(Na_2_PO_3_)_6_
Reference	-	-	0.35	1	0.01	0.005
A1	PVA	0.005	0.35	1	0.01	0.005
A2	PVA	0.01	0.35	1	0.01	0.005
A3	PVA	0.02	0.35	1	0.01	0.005
B1	PAA-Na	0.005	0.35	1	0.01	0.005
B2	PAA-Na	0.01	0.35	1	0.01	0.005
B3	PAA-Na	0.02	0.35	1	0.01	0.005
C1	CPAM	0.005	0.35	1	0.01	0.005
C2	CPAM	0.01	0.35	1	0.01	0.005
C3	CPAM	0.02	0.35	1	0.01	0.005

**Table 3 materials-17-00734-t003:** Mixture proportions for preparing samples in large water–cement ratios.

Alkali Content (%)	NaOH (g)	Slag (g)	Water (g)
2	5.36	20	70
4	10.84	20	70
6	16.12	20	70
8	21.50	20	70

**Table 4 materials-17-00734-t004:** Polymerization distribution and main chain length of slag-based geopolymers prepared with NaOH activators with different alkali contents (28 d): (a) 2% NaOH; (b) 4% NaOH; (c) 6% NaOH; (d) 8% NaOH.

Sample	Q^n^ Cumulative Strength (I/%)							
	Q^0^	Q^1^(1Al)	Q^1^(0Al)	Q^2^(1Al)	Q^2^(0Al)	Q^3^(1Al)	Q^3^(0Al)	MCL
(a)	18.91	22.73	20.01	24.23	10.37	-	3.75	6.12
(b)	16.96	23.25	20.59	25.05	9.09	3.67	1.39	6.51
(c)	14.06	25.2	21.62	27.52	9.44	2.10	0.09	6.69
(d)	15.30	26.56	24.19	26.86	7.09	-	-	5.91

**Table 5 materials-17-00734-t005:** Degree of polymerization distribution and main chain length of slag-based geopolymers prepared with different modulus activators (28 d): (a) 6%−0.8; (b) 6%−1.2; (c) 6%−1.6; (d) 6%−2.0.

Sample	Q^n^ Cumulative Strength (I/%)							
	Q^0^	Q^1^(1Al)	Q^1^(0Al)	Q^2^(1Al)	Q^2^(0Al)	Q^3^(1Al)	Q^3^(0Al)	MCL
(a)	16.45	15.44	21.33	29.53	11.60	3.68	1.97	6.81
(b)	13.67	17.31	18.85	26.54	15.22	5.15	2.96	7.83
(c)	7.64	19.81	17.59	24.44	19.33	6.18	5.01	8.36
(d)	6.53	21.89	25.24	-	36.32	-	10.02	7.49

**Table 6 materials-17-00734-t006:** Effect of polymer type and dosage on compressive and flexural strengths of paste.

	Compressive Strength (MPa)			Flexural Strength (MPa)		
	3 d	7 d	28 d	3 d	7 d	28 d
Reference	71.8	82	91	5.1	6.4	7.5
A1	77.7	89.6	98.3	6.9	8.3	9.2
A2	73.1	83.4	90.3	6	7.2	8.3
A3	60	70.7	76.2	5.3	6.1	7.3
B1	65	70.2	78.3	4.6	5.2	6.2
B2	62.4	64.3	71.5	4.4	4.9	5.8
B3	58.9	60	62.5	3.7	4.5	5.3
C1	63	65.6	70.9	4.8	5.9	6.6
C2	57.1	59.7	63.2	4.1	5.2	6
C3	51.7	57.8	60.1	4.4	4.7	5.5

**Table 7 materials-17-00734-t007:** Polymerization distribution, main chain length, and reaction degree of paste at 28 d.

PVA Content	Q^n^ Cumulative Strength (I/%)								
	Q^0^	Q^1^(1Al)	Q^1^(0Al)	Q^2^(1Al)	Q^2^(0Al)	Q^3^(1Al)	Q^3^(0Al)	MCL	α
0%	7.64	19.81	17.59	24.44	19.33	6.18	5.01	8.36	82.85%
0.5%	13.98	17.77	19.22	29.31	18.85	-	0.87	8.70	73.14%

**Table 8 materials-17-00734-t008:** Polymerization distribution of pastes prepared by pressure-mixing process under different curing regimes at 28 d.

Content Curing Condition	Q^n^ Cumulative Strength (I/%)						
	Q^0^	Q^1^(1Al)	Q^1^(0Al)	Q^2^(1Al)	Q^2^(0Al)	Q^3^(1Al)	Q^3^(0Al)
1%	14.59	21.39	24.57	17.79	17.48	2.31	1.87
3%	19.99	16.63	22.05	18.98	27.28	-	5.07
5%	21.73	18.88	19.45	20.20	15.35	-	4.39
3% dry	9.34	18.39	20.85	25.14	15.80	-	10.03
3% steam	10.07	8.99	16.91	19.50	33.72	-	10.51

## Data Availability

Data are contained within the article.

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
