# Peer review of "The Mechanical Performance and Reaction Mechanism of Slag-Based Organic–Inorganic Composite Geopolymers"

_materials, 2024, doi:10.3390/ma17030734_

Round 1

Reviewer 1 Report

Comments and Suggestions for Authors

Reviewing Manuscript: materials-2754865

 Mechanical Performance and Reaction Mechanism of 2 Slag-based Organic-inorganic Composite Cementitious 3 Materials

Referee Comments:

The whole analysis is of interest in the areas of composite materials, polymers, and geopolymers. All the analyses are significant for showing the behavior of the composites.

1.

It is not appropriate the careless error in leaving the instructions as the first and last paragraphs in the Introduction section.

Delete the following paragraphs:

[lines 26-34]

 The introduction should briefly place the study in a broad context and highlight why it is important. It should define the purpose of the work and its significance. The current state of the research field should be carefully reviewed, and key publications cited. Please highlight controversial and diverging hypotheses when necessary. Finally, briefly mention the main aim of the work and highlight the principal conclusions. As far as possible, please keep the introduction comprehensible to scientists outside your particular field of research. References should be numbered in order of appearance and indicated by a numeral or numerals in square brackets—e.g., [1] or [2,3], or [4–6]. See the end of the document for further details on references.

[line 95]

All figures and tables should be cited in the main text as Figure 1, Table 1, etc.

Similarly, the first paragraph in section 2.2 are the instructions.

[lines 121-123]

This section may be divided by subheadings. It should provide a concise and precise description of the experimental results, their interpretation, as well as the experimental conclusions that can be drawn.

2. The whole first paragraph of the Conclusions section comes from the reference [https://doi.org/10.1016/j.conbuildmat.2018.02.031]. It is recommended to write it again or delete it. Similarly, some sentences in the points 1 and 2 of this section.

3. There are other writing errors that should be amended, such as putting value and unit together “2min”, “4min”, “80°C”.

4. Section 3.1.3 could include a couple of references for the vibrational bands mentioned. Indeed, there is information used from the references (https://doi.org/10.1021/cm990475t).

5. Figure 7 could be shown better with some changes.

6. Figure 9 caption requires a period at the end.

The manuscript has a good organization and the analyses were discussed properly.

Comments on the Quality of English Language

The English grammar is adequate, but still there are some issues.

Reviewer 2 Report

Comments and Suggestions for Authors

The paper provides insight into a use of three different organic polymers on the mechanical properties and reaction mechanism of composite cementitious material. The paper has an interesting premise, however there are some issues that I would like Authors to adress: 

1. There are descriptions of formatting rules of the journal left in the text

2. The formulas 2-1 to 2-4 have no source/reference. 

3. There are no information about the chosen polymers in materials section outside of their name. 

4. Table 2 is not clear - it is described as mixture proportions in %, but no value is higher than 1%; those are also not mortar proportions, missing sand, cement etc. Moreover, the same paragraph that references tab. 2 references tab.3. which is located far from this point, which also do not have all the data neccessary to recognize what is the composition. Please remake the materials section to better show what is the composition, as currently it is very unclear. 

5. Text refers to the mix as mortar and paste - which was used? 

6. Why strength was not tested according to any standards, for example for cement? If they were, pease provide reference, if not - it would be beneficial to add an explanation of the procedure. The strength testing is not well described. 

7. Two production methods were used - it would be beneficial to explain why. 

8. The description of methods is lacking description of what kind of sample was used for which test and how many samples were used for one test. 

9. The discussion is in places very chaotic and sometimes the conclusions in  discussion do not correspond well to the text, e.g. "The result indicate that the mixing-suppression process eliminates the negative effect on the strength of the polymer, significantly improves the flexural strength." does not fully connect to the rest of text in 3.3.1. 

10. Discussion may benefit from references to the existing state of the art. 

Comments on the Quality of English Language

Many sentences are very hard to understand, and are not fully understandable, both due to gramatical errors and missing words e.g.: 

"It is known from the formula (2-1) to (2-4), which is known as high quality slag" 

"Sodium inorganic water reduction (solid analysis pure, 118 Tianjin Fu Chen Chemical Reagent Factory) was selected." 

"Among this experiment, sodium hydroxide solution as the alkali activator."

"As Figure 3 198 shown, it can be seen that the majority of compression and flexural strength happens 199 when alkali content is 6%."

"There is a large diffuse band at about 1003-1035 cm-1 indicating the production of the 228 amorphous consisting of Si-Al polytetrahedras with a lack of periodically"

Moreover, text sometimes uses wrong words: "FTIR analysis was detected in this study" from what I understand, be "FTIR analysis was conducted in this study"

Reviewer 3 Report

Comments and Suggestions for Authors

This work deals with the design of different organic-inorganic hybrid cementitious materials based on slag derivative geopolymers mixed with three water-soluble organic polymer:, namely the cation polyacrylamides (CPAM), the polyvinyl alcohol (PVA) and the polyacrylic acid (PAA). The resulting materials were characterized by different methods, with particular focus on their mechanical properties, morphological and structural features. Overall, the paper is well presented and its topic falls within the scope of journal. However, there are some major ravisions that need to be addressed before publication.

1.Please reshape the paragraph concerning the materials and methods. All the specific properties of starting reagents (such as their purity degree) should be reported.

2.How were the mechanical properties measured? Which instrumen was used and what conditions for experiments?

3. In paragraph 2.4.4. the type of SEM microscope, the working distance, the electron beam energy and the other important experimental paramenters are missing. Please add and check also the other methods.

4. In IR graph, the y ax is lacking of unit. Please add it (maybe a.u. as arbitrary units).

5. Why the incorporation of PVA until 3% increased the strength of the materials and the further increase to 5% did not had good effect on them? Please provide some explanation.

6.The authors often refer to the treatment or alkali activation of the starting slag by using NaOH. They discussed the dissolution of silicon or silicon based ions. It would be interesting to compare results with literature, especially to recent works about the pretreatment of silicon containing or silica(-te) based materials to explain the breaking of certain bonds (please cite: https://doi.org/10.1016/j.clay.2022.106813).

7.It is strongly recommended to revise and check the whole manuscript, as they are some parts that seem to need to be removed (i.g. lines 25/33 and 121/123). Also, it is recommended to have the manuscript read and corrected by an english native speaker.

8.Please enrich the introduction with the most recent literature reporting the using of geopolymers based materials for building or coating materials (see and cite: https://doi.org/10.1016/j.clay.2022.106786, https://doi.org/10.3390/ma17010142,).

Comments on the Quality of English Language

extensive editing needed 

Round 2

Reviewer 2 Report

Comments and Suggestions for Authors

The Author made many changes improving the paper. I would only inquire about the answer to comment 7: 
"Comments 7: Two production methods were used it would be beneficial to explain why.

Response: Thank you for the comment. T wo production methods were used in the study. One was for strength test and the other was for........." 

I believe something was lost in the editing process. 

Reviewer 3 Report

Comments and Suggestions for Authors

The authors addressed all comments and suggestions. The paper can now be accepted for publication. 

Comments on the Quality of English Language

minor editing 
